# Predictability of Astigmatism Correction by Arcuate Incisions with a Femtosecond Laser Using the Gaussian Approximation Calculation

**DOI:** 10.3390/mi14051009

**Published:** 2023-05-07

**Authors:** Isabel Llopis Sanmillan, Gabriele Thumann, Martina Kropp, Zeljka Cvejic, Bojan Pajic

**Affiliations:** 1Eye Clinic ORASIS, Swiss Eye Research Foundation, 5734 Reinach, Switzerland; llopis.isabel@gmail.com; 2Division of Ophthalmology, Department of Clinical Neurosciences, Geneva University Hospitals, 1205 Geneva, Switzerland; gabriele.thumann@hcuge.ch (G.T.); martina.kropp@unige.ch (M.K.); 3Experimental Ophthalmology, University of Geneva, 1205 Geneva, Switzerland; 4Faculty of Sciences, Department of Physics, University of Novi Sad, Trg Dositeja Obradovica 4, 21000 Novi Sad, Serbia; zeljka.cvejic@df.uns.ac.rs; 5Faculty of Medicine of the Military Medical Academy, University of Defense, 11000 Belgrade, Serbia

**Keywords:** femtosecond laser, arcuate incisions, customised surgery calculation

## Abstract

Planning astigmatic correction is a complex task. Biomechanical simulation models are useful for predicting the effects of the physical procedure on the cornea. Algorithms based on these models allow preoperative planning and simulate the outcome of patient-specific treatment. The objective of this study was to develop a customised optimisation algorithm and determine the predictability of astigmatism correction by femtosecond laser arcuate incisions. In this study, biomechanical models and Gaussian approximation curve calculations were used for surgical planning. Thirty-four eyes with mild astigmatism were included, and corneal topographies were evaluated before and after femtosecond laser-assisted cataract surgery with arcuate incisions. The follow-up time was up to 6 weeks. Retrospective data showed a significant reduction in postoperative astigmatism. A total of 79.4% showed a postoperative astigmatic value less than 1 D. Clinical refraction was significantly reduced from −1.39 ± 0.79 D preoperatively to −0.86 ± 0.67 D postoperatively (*p* 0.02). A positive reduction in topographic astigmatism was also observed (*p* < 0.00). The best-corrected visual acuity increased postoperatively (*p* < 0.001). We can conclude that customised simulations based on corneal biomechanics are a valuable tool for correcting mild astigmatism with corneal incisions in cataract surgery to improve postoperative visual outcomes.

## 1. Introduction

Refractive errors, followed by cataracts, are the most common causes of visual impairment [1]. Among the refractive errors, astigmatism is the most common error in adults worldwide at 40.4%, according to a meta-analysis published in 2018 with a sample size of 122,436 adult participants [2].

Cataract surgery is one of the most commonly performed refractive procedures. Most patients who are scheduled for cataract surgery have astigmatism. More than 50% have astigmatism greater than 1 dioptre (D), and in the vast majority (64.4%), the prevalence of corneal astigmatism is between 0.25 and 1.25 D [3,4]. Postoperatively, corneal astigmatism persists [5,6]. It has been proven that uncorrected astigmatism of 1.00 D or more has a negative effect on visual acuity and leads to a deterioration of visual function and a higher best-corrected visual impairment [7,8].

Simultaneous treatment and a significant reduction in astigmatism are, therefore, advisable to achieve optimal visual acuity and visual quality after cataract surgery. By using the femtosecond laser, an already existing astigmatism can be reduced in the same procedure and, thus, better visual acuity can be achieved postoperatively.

Currently, there are several treatment options for corneal astigmatism during cataract surgery: implantation of a toric intraocular lens and incisional treatments of the cornea. Arcuate or astigmatic keratotomy, femtosecond laser-assisted arcuate keratotomy and limbal relaxing incisions (LRI) fall within the spectrum of incisional astigmatism treatments.

Studies have shown that femtosecond laser-assisted astigmatic keratotomy is a good and effective method for reducing corneal and, therefore, refractive astigmatism in cataract surgery [9,10]. This astigmatic correction was found to have a good safety profile with very good refractive topographic stability over time without a significant change in endothelial cell density [11,12]. Further studies have stated that corneal astigmatism after arcuate keratotomy (AK) during femtosecond-assisted phacoemulsification in patients with a mean preoperative astigmatism of less than 1.50 D was significantly reduced postoperatively and remained stable at 2 months, 2 years and 5 years postoperatively [13,14]. By making arcuate incisions with the femtosecond laser during cataract surgery, astigmatism correction and, thus, better uncorrected visual acuity can be achieved postoperatively for mild astigmatism levels.

Surgical procedures on the anterior segment of the eye alter the biomechanical properties, viscous and elastic properties and balance and shape of the cornea [15]. Variations in the corneal shape and biomechanics significantly affect surgical outcomes. Therefore, biomechanical simulation models are useful to predict the effects of the physical procedure on the cornea. They allow patient-specific preoperative planning, reduce outliers, improve safety and achieve optimal postoperative outcomes [16]. 

There are several nomograms and variations of nomograms that determine the parameters for AK, such as the angular length, position, depth, optical zone diameter and distance from the visual axis where the incisions should be made [17,18,19]. To date, no optimal methodology or nomogram has been developed that is specifically and universally applicable to the use of the femtosecond laser in cataract surgery.

In many cases, the treatment of astigmatism with arcuate incisions is based on a nomogram based on the statistical data of a large number of patients. Detailed topographic data, as well as pachymetry or aberration, were not considered. The aim of the present study was to determine the numerical predictability of astigmatism correction by arcuate incisions with a femtosecond laser using an optimised algorithm in a customised application, i.e., a Gaussian approximation derived from corneal biomechanics and a desirable reduction in postoperative astigmatism. Based on the results, we hope to develop a new clinical approach and possibly improve current treatment goals and strategies.

## 2. Materials and Methods

### 2.1. Surgery Technique with Femtosecond Laser of the Arcuate Incisions

A retrospective case series study was conducted. Thirty-four eyes of twenty-five patients were included in the study. The mean age of the patients was 72 ± 7.2 years (range 59–90 years). Femtosecond laser-assisted cataract surgery (FLACS) was performed with arcuate incisions in the same treatment algorithm. Postoperatively, the clinical status of the patients was assessed on day 1, day 3 and after 2, 4 and 6 weeks. Corneal topography was performed preoperatively and 6 weeks postoperatively.

An IOL master (Carl Zeiss Jena, Jena, Germany) was used for biometry in the context of cataract surgery. Optical biometry is based on the principle of coherence interferometry. This measurement was used to calculate the intraocular lens power, using the SRK-T formula to estimate the target refraction.

For the arcuate incision, the study used an LDV Z8 femtosecond laser (Ziemer Instruments, Port, Switzerland), an oscillating ytterbium-doped YAG (Yb:YAG) laser operating at a wavelength of 1030 nm. A suction ring is used to fix the bulb before the laser head of the femtosecond laser is clicked into place in a liquid interface. The pulse energy is in the range of nJ with a pulse length of 200 fs. There is an adaptive optical system to focus optimally at each tissue level. The arcuate incisions were performed as part of the cataract surgery, and in the treatment algorithm of the femtosecond laser, the arcuate incisions were performed first and then the capsulotomy was performed, followed by the lens fragmentation and clear corneal incision. The femtosecond laser used has a large optical numerical aperture, which allows a small precise laser spot. Due to a high pulse rate in the MHz range, the laser spots applied allowed overlapping, which enabled a smooth incision without tissue bridges. Optical coherence tomography (OCT) (Ziemer Instruments, Port, Switzerland) is an extremely important imaging technique for the application of the femtosecond laser. It provides high-resolution, non-invasive optical images of every interface of all the structures of the anterior bulbar segment. It is directly coupled to the femtosecond laser so that it operates like a navigation system. The incision patterns in the cornea are preoperatively defined and entered into the computer, although intraoperative fine-tuning of the incision patterns can be made based on the OCT images if required. The appropriate tools can be adjusted on the screen. The OCT ensures that all the cuts with the femtosecond laser are made very precisely at the right place and in the right shape in the cornea.

The treatment algorithm of the femtosecond laser is as follows. After the OCT imaging procedure and the definition of the incision in the images, the femtosecond laser begins with the application of the arcuate incisions. In the second step, the capsulotomy is performed very precisely with an application diameter of 5 mm in all the patient eyes. In the third step, the lens is fragmented with the femtosecond laser. In our surgical application, six spider pieces were cut with a central cone, resulting in twelve pieces in the end. The pieces can be safely removed from the capsular bag with the phacoemulsification tip, which had a diameter of 2.2 mm in the study. In the fourth and final step, clear corneal incisions are made, with the diameter of the main incision at 2.3 mm and the two side ports at 0.8 mm. The surgery is then continued with phacoemulsification until the core parts of the lens are evacuated. The phacoemulsification device used was the Catharex 3 (Oertli Instrumente AG, Berneck, Switzerland). The cortical parts of the lens are removed by irrigation and aspiration and, if necessary, the capsule is polished. As a further step of the procedure, a folded IOL is implanted through the main incision into the capsular bag under the control of viscoelastic (Pe-th-Visco 2.0% 2 mL) (ALBOMED GmbH, Schwarzenbruck, Germany), which keeps the anterior chamber stable during this procedure. Finally, the viscoelastic is completely removed by irrigation and aspiration, and the antibiotic Cefuroxime (Aprokam, Thea Pharma, Clermont-Ferrand, France) is applied intraoperatively.

Postoperatively, all the patients received Tobradex and Nevanac 0.1% drops (Alcon Laboratories, Inc., Fort Worth, TX, USA) for 4 weeks. Remotely, hyaluronic acid 0.15% drops (Thea Pharma, Clermont-Ferrand, France) were applied for at least 2 months for better postoperative corneal surface lubrication.

One or two symmetrical incisions were made on the steep corneal axis depending on the calculation recommendation. Due to the known coupling effect, the flattening of the incision on the steep meridian is accompanied by a steepening on the meridian 90° away. The steep corneal axis was determined by topography. A total of 17 left and 17 right eyes were included. All the included patients had mild to moderate astigmatism. The inclusion criteria were patients with indicated cataract surgery and mild corneal astigmatism before surgery of more than 0.60 D and less than 2.50 D. Patients with pronounced corneal astigmatism with values above 2.50 D, other corneal diseases, ocular trauma or previous corneal or intraocular surgery were excluded. The data were collected continuously from June 2018 to December 2021.

### 2.2. Calculation and Application of the Treatment Parameters of the Arcuate Incisions

The shapes of the corneas were recorded with a Ziemer Galilei G4 Dual Scheimpflug Analyser (Ziemer Ophthalmic Systems, Port, Switzerland) and a Pentacam HR Scheimpflug camera (Oculus, Optikgeräte GmbH, Heidelberg, Germany). Both topography devices, combined with a rotating Scheimpflug camera, allow accurate measurements of the anterior and posterior surfaces and analysis of the centre of the cornea. With the Galilei G4 measurement, analysis of Placido rings is used in addition to the Scheimpflug recording.

Biomechanical modelling and Gaussian approximation curve calculations were used for surgical planning. By creating virtual biomechanical simulations, an identical virtual copy of the patient’s cornea can be created based on corneal topography measurements. In this way, it is possible to simulate and evaluate the patient-specific treatment outcome already in the planning phase of the procedure. The topography values were imported, and the intervention was planned and optimised virtually. From the selected topography data, the main axes of astigmatism were determined, and algorithms were used to plan the procedure and minimise the risks in advance. These measurements helped to adjust the incision pattern and calculate the length of the incisions for each cornea to achieve more accurate astigmatism reduction and better results.

It is known that corneal tissue has non-linear elastic properties. In addition, corneal tissue is almost incompressible. Both in the plane and across the thickness profile of the cornea, it is highly inhomogeneous and exhibits a high degree of anisotropy.

For our work, we used existing biomechanical models [20,21,22]. In a so-called strain energy function, the model used additive terms to describe the biomechanical properties of the soft-biological tissue.
(1)$$\psi =U+\psi_{m}\;\left[C_{10}\right]+\frac{1}{\pi }\;\int \Phi \;(\psi_{f1}\left[{ϒ}_{m},\;{µ}_{m\;}\right]+\psi_{f2}\;\left[{ϒ}_{k},\;{µ}_{k}\right])\;d\theta$$

The polynomial material function models the collagen fibres or cross-links. The penalty function models incompressibility. The matrix and penalty function are modelled with Neo Hook representing the tissue matrix. For the distribution of collagen fibres, a mathematical distribution function defines a realistic probability [23,24,25,26], with individual weights assigned to each fibre direction. The distribution is not only a function of the direction but also a function of the corneal depth. By inverse fitting the above mathematical function (Equation (1)), the material parameters were determined for several sets of age-dependent experimental data. *ψ* is the energy required to deform the material. The right-hand side expresses the deformation. *U* is a penalty function for the volume changes, and *ψ_m_* is a neo-Hooke’s material and represents the tissue matrix. *ψ*_*f*1_ and *ψ*_*f*2_ are modified Ogden materials that model the main collagen fibres and the branched fibres (cross-links) and add shear stiffness to the material definition. The probability distribution function, Φ, defines the set of collagen fibres [27,28,29].

In a four-step process, a patient-specific finite element model was created for each eye. The finite element method (FEM) is a mathematical approach to solving complex mechanical problems by dividing a structure, such as a cornea, into small elements of finite size. The FEM has also been used in other studies simulating other corneal surgical procedures [30,31]. 

The cornea was measured using Scheimpflug topography, and, in particular, the pachymetry was recorded. These data were each converted into a patient-specific finite element mesh consisting of 35,000 elements and 44,000 nodes. An iterative algorithm was used to calculate the individual initial stress distribution of the cornea, which enables an individualised analysis [22]. Since the tension-free shape of the cornea is not known at the time of the Scheimpflug measurement because eye pressure is naturally present, this described analysis must be used. The intrastromal arcuate keratotomy operations were simulated individually in ANSYS solver (ANSYS Inc., Canonsburg, PA, USA) by cutting nodes in the finite element mesh at the exact locations where the arcuate cuts were made clinically with the laser system. The resulting anterior and posterior surfaces of the finite element mesh were then compared to the corresponding postoperative surfaces measured with Scheimpflug topography.

The following parameters must be defined, calculated and then entered into the femtosecond laser for the arcuate incision before starting the treatment. First, the diameter of the arcuate incision must be entered and then the ARC angle is entered, followed by the position of the incisions (Figure 1).

Intraoperatively, the online OCT image of the calculated arcuate incisions is shown. Preoperatively, the angle at the limbus is marked so that the alignment intraoperatively can be even more precise, especially to consider a possible cyclorotation (Figure 2).

The last relevant parameter calculated is the depth of the corneal incision. It is given as a percentage and refers to the total thickness of the cornea in the area of the planned incision. Therefore, the precision of the OCT is very important in connection with the cutting function of the femtosecond laser (Figure 3).

The diameter of the optical zone in the asymmetric, single arcuate incision group was 8.7 ± 1.42 mm. The mean aperture angle was 36.0 ± 10.23°. The corneal incision depth of the arcuate incision was 82.3 ± 7.61%. The percentage refers to the complete corneal thickness in the area of the incision. The diameter of the symmetrical application of the arcuate incision group, i.e., with symmetrical incisions, was 9.1 ± 0.46 mm. The average opening angle was 34.7 ± 7.61° with a corneal incision depth of 82.7 ± 3.89%. The incision is made 90° to the cornea and is continuous, i.e., the femtosecond laser cuts not only the stromal but also through the epithelium to the surface of the cornea.

As an example, an eye is shown after the femtosecond incisions have been made. Next to the spider lens fragmentations, a corneal arcuate incision can be seen that was previously calculated. Smaller bubbles (CO_2_) are visible due to the cutting process with the femtosecond laser (Figure 4).

Statistical calculations and data analysis were performed using IBM SPSS Statistics software version 22.0 (IBM Corp., Armonk, NY, USA). A *p*-value of less than 0.05 was considered statistically significant. The Kolmogorov–Smirnov and Shapiro–Wilk tests were used to test for normal distribution. The criterion for a normal distribution was if *p* > 0.05. The paired *t*-test was used for parametric data sets. The Wilcoxon test was used for non-parametric data sets for the calculation. The data collection was anonymous, and the clinical trial was approved by the Ethics Committee of Northwestern and Central Switzerland-EKNZ (2022-01726).

## 3. Results

### 3.1. Demographics

The study included 34 eyes of 25 patients who underwent arcuate incisions in FLACS. The mean age was 72 ± 7.1 years at the time of surgery. One arcuate keratotomy incision (AK) was performed in 29.4% of patients and two AKs were performed in 70.6% of patients.

The data were analysed using the Kolmogorov–Smirnov test and the Shapiro–Wilk test to determine whether or not there was a normal distribution. Based on these results, the parametric and non-parametric tests were further selected.

### 3.2. Clinical Outcomes of Astigmatism Correction

Postoperative astigmatism determined by clinical refraction was reduced from preoperative 1.39 ± 0.79 D to postoperative 0.86 ± 0.67 D, which was significant (*p* = 0.02). 

The mean topographic astigmatism was 1.27 ± 0.38 D (range 0.69 to 2.04 D) preoperatively. The distribution of the individual values is shown in the double-angle plot (Figure 5).

Postoperatively, there was a significant reduction in astigmatism to 0.86 ± 0.32 D (range 0.16 to 1.6 D), which was significant (*p* < 0.001). The distribution of the individual values can be seen in the double-angle plot (Figure 6). In 79.4% of all postoperative topographic measurements, the astigmatism value was below 1 D.

The preoperative calculated astigmatism was 0.42 ± 0.05 D. The individual calculated postoperative astigmatism can be seen in the double angle plot (Figure 7). 

Both the preoperative values and the achieved postoperative values were significantly different to the calculated values (*p* < 0.001).

The procedures were planned before surgery using patient-specific AK planning. Target-induced astigmatism (TIA) is astigmatism in power and angle that is to be obtained by our customised calculation and that is targeted by the procedure. This is a calculated target value only. Surgically induced astigmatism (SIA) refers to effective astigmatism in power and angle obtained clinically and measured after arcuate incisions. From the two graphs in Figure 8, the outliers and the distribution of the difference vector can be seen, including the magnitude and the axis. The comparison of the planned astigmatism correction and the effective result shows an undercorrection. However, no overcorrection was observed in any case.

There was no correlation between topographic preoperative and postoperative astigmatism. This means that preoperative astigmatism has no influence on how well the astigmatism can be corrected (R^2^ = 0.0658). There was no correlation between the age of the patient and postoperative astigmatism (R^2^ = 0.0019). The age of the patient has no influence on how well the astigmatism can be corrected. There was no significant difference in the postoperative astigmatism values between the eyes where incisions for astigmatism correction were performed asymmetrically, i.e., with one arcuate incision (0.83 ± 0.34 D) versus a symmetrical incision, i.e., two arcuate intracorneal incisions were performed (0.87 ± 0.32 D) (*p* > 0.05).

### 3.3. Visual Acuity and Target Refraction

The best-corrected visual acuity (BCVA) was 0.57 ± 0.16 preoperatively and increased to 0.93 ± 0.12 postoperatively, which was significant (*p* < 0.001) (Figure 9).

The biometrically calculated target spherical equivalent refraction was −0.46 ± 0.45 D. Postoperatively, a spherical equivalent refraction of −0.63 ± 0.41 D was clinically achieved. Although there was a slight myopic shift, the difference was not significant (*p* = 0.13).

## 4. Discussion

In our study, we show a new approach to optimise the parameters for performing arcuate incisions with femtosecond lasers without using nomograms based on a customised calculation. In particular, the significantly improved analysis and reproducibility of topography measurements in combination with the highly precise application of the femtosecond laser in the formation of arcuate cuts can potentially lead to better results in terms of astigmatism reduction. It has already been shown in one study that for accurate simulations, it is necessary to determine the existing strain in soft tissues and their previous stress-free configuration. Thus, a method for determining the stress-free configuration using the iterative finite element method (FEM) approach was proposed. It proved to be an effective technique for rapid prediction of the stress-free shape of the cornea. By creating a finite element mesh of the tissue in its relaxed configuration, its deformed patient-specific configuration could be determined by applying intraocular pressure, which allowed for good simulation in refractive surgery [32]. We also followed this customised path in our work.

The aim of this retrospective study was to evaluate the efficacy of intrastromal arcuate incisions with a femtosecond laser using a calculated customised approximation curve based on corneal biomechanics. Several studies have already reported that the femtosecond laser is a safe, reliable method with well-reproducible arcuate intracorneal incisions for the correction of astigmatism, but we hypothesised that customised preoperative biomechanical modelling can further improve the surgical outcome [9,10]. Further prospective studies with a control group need to be performed in order to better assess the potential of the method.

A randomised case–control study compared the efficacy of femtosecond laser-induced arcuate keratotomies with manual limbal relaxation incisions. The results showed that the femtosecond arcuate keratotomy group achieved a higher correction index and lower postoperative cylinder than manual limbal relaxation incisions [33]. Another study [33] demonstrated a significant reduction in both postoperative mean refractive astigmatism and topographic astigmatism (1.50 ± 0.47 to 0.63 ± 0.34 D) (*p* < 0.001). High patient satisfaction could be stated in this work [12]. Our study also shows a significant reduction in the refractive cylinder and a remarkable reduction in postoperative topographic astigmatism (1.27 ± 0.38 D to 0.86 ± 0.32 D), which is in very good agreement with the studies mentioned above.

In another study, the uncorrected and corrected visual acuity after arcuate incisions with the femtosecond laser resulted in high patient satisfaction [34]. In another study of intrastromal arcuate incisions with a femtosecond laser, 85.7% of the patients had postoperative astigmatism of less than 1 D. Here, the surgically induced astigmatism (SIA) was below the predicted target-induced astigmatism (TIA) [34]. In our study, 79.4% of topographic astigmatisms were below 1 D, which is slightly lower than in the cited paper. However, we also obtained a difference between the SIA and TIA after surgical intervention. Other studies also showed that postoperative undercorrection is seen with cataract surgery combined with astigmatism correction through arcuate incisions [33,35].

Until recently, only nomograms were used to conduct the arcuate incisions. All these nomograms were based on empirically collected data. The higher the number of patient data included in the nomograms, the more accurate their calculations. Although, on average, these nomograms were very close to the target value, it was not uncommon for overcorrections to occur in some treated corneas [17,36]. In our study, we found an undercorrection but did not observe an overcorrection in any of the cases, which can certainly be seen as an advantage of the customised calculatory treatment method for arcuate incisions. 

In another published paper, a customised planning tool was used for computer modelling to simulate the biomechanics of the cornea. With this method, an astigmatism reduction from 1.24 ± 0.43 D preoperatively to 0.80 ± 0.50 postoperatively could be achieved, but intracorneal arcuate incisions were performed [37]. In our work, we arrived at analogous values with a similar calculatory approach, although we did not quite achieve the calculated values clinically. A previous study showed that customised simulations of corneal arcuate incisions came close to the refractive clinical results [30]. This supports our study, which uses a patient-specific, customised approach as the basis for treatment.

The best-corrected visual acuity (BCVA) improved significantly from 0.57 ± 0.16 to 0.93 ± 0.12 after FLACS and arcuate incisions. On the one hand, this shows the improvement in visual acuity that we expect after an IOL change during cataract surgery, and, on the other hand, it gives us an important indication that the arcuate incisions have no negative influence on the BCVA.

The limitations of our study were that it was retrospective, only a limited number of patients were available and there was no control group. 

In summary, we were able to achieve a significant reduction in corneal astigmatism through the customised calculation of corneal arcuate incisions, which is supported by other studies. However, it should be mentioned that, on average, we did not achieve the calculated and targeted astigmatism correction. Compared to studies in which nomograms were used as the basis for the treatment, treatment according to the protocol in our study produced no overcorrections, which are not well tolerated by patients. 

## 5. Conclusions

The results of this study show that topographic and refractive astigmatism reduction can be achieved using corneal biomechanical calculations and femtosecond laser-assisted arcuate incisions. The femtosecond laser is a highly accurate, reliable and safe tool for astigmatism correction as it can create precise corneal incisions at a variety of depths and orientations. Its use in conjunction with the calculation of the specific biomechanics of the patient’s cornea allows for an optimal evaluation of the treatment outcome in the planning phase and excellent results in the postoperative correction of astigmatism. Research on the effects of corneal biomechanical behaviour is a growing field of interest. Further prospective studies are needed to evaluate the long-term outcomes.

## Figures and Tables

**Figure 1 micromachines-14-01009-f001:**
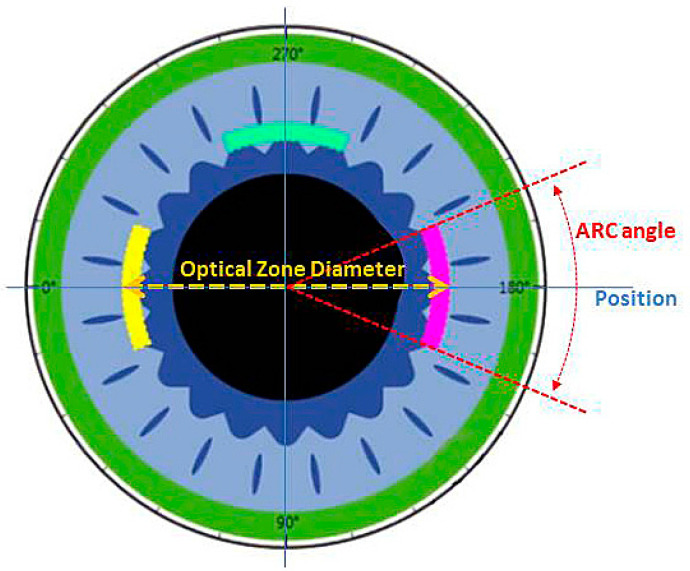
Schematic representation of the cornea with the cutting parameters of the arcuate incisions, particularly, the optical diameter, cutting angle and cutting position.

**Figure 2 micromachines-14-01009-f002:**
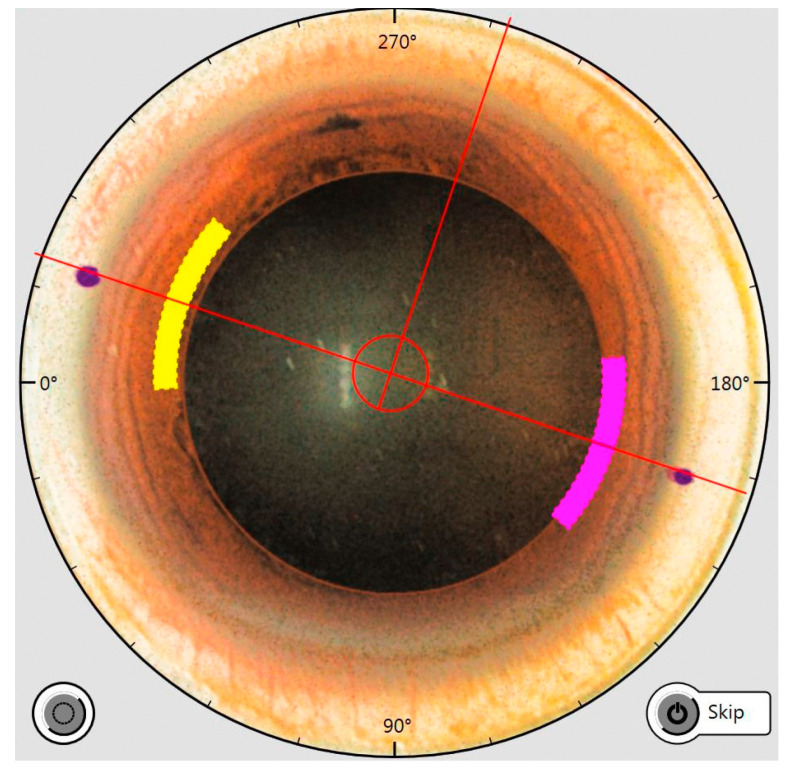
Intraoperative OCT image showing the cutting parameters of the arcuate incisions, i.e., the optical diameter, the cutting angle and the position. Limbally, two markers are visible that were placed before the procedure in the vertical patient orientation, which determines the position of the arcuate incisions.

**Figure 3 micromachines-14-01009-f003:**
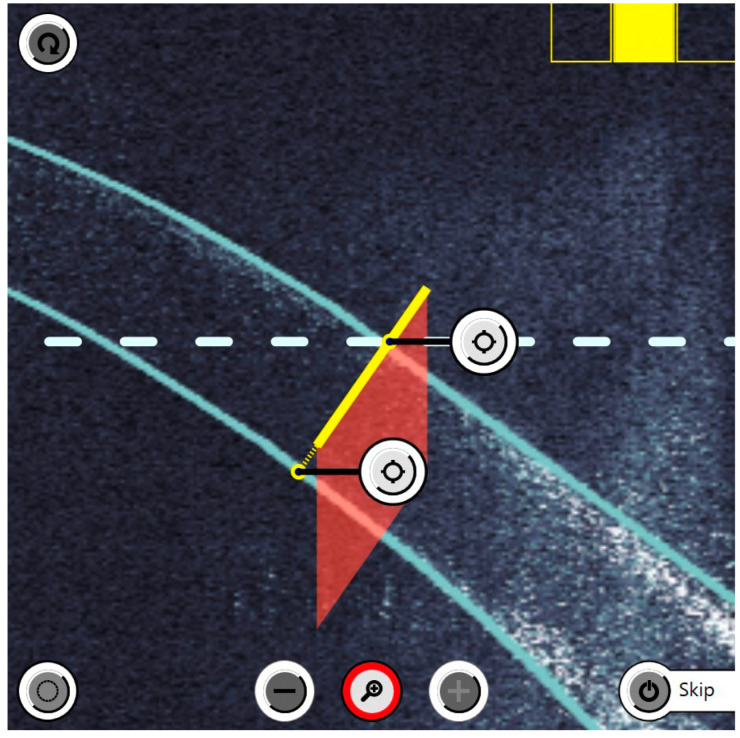
Intraoperative OCT imaging shows the corneal thickness so that the depth of the incision can be set very precisely. The yellow continuous line represents the planned incision. This visualisation option significantly improves the safety of the treatment.

**Figure 4 micromachines-14-01009-f004:**
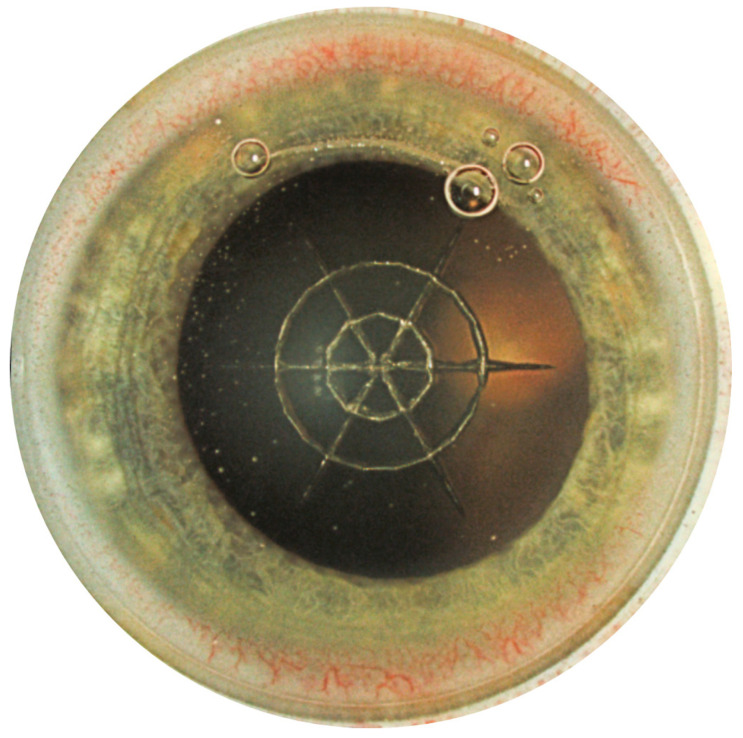
After the incision, this is how the eye presents itself intraoperatively. The arcuate incision can be seen at the top of the image between two bubbles. The lens fragmentation can be seen in the centre with six radial parts and two concentric cuts.

**Figure 5 micromachines-14-01009-f005:**
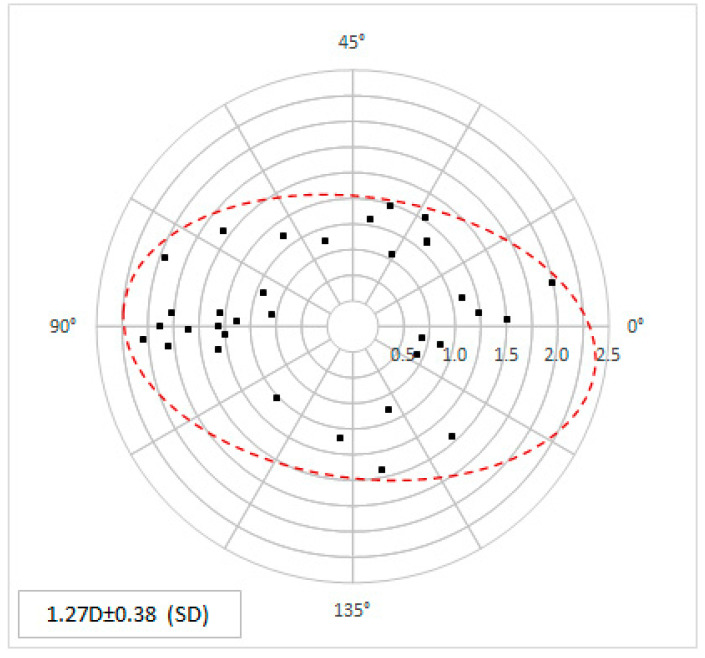
The preoperative topographic astigmatism values in power and angle are shown as double-angle plots. Each point represents the individual corneal data.

**Figure 6 micromachines-14-01009-f006:**
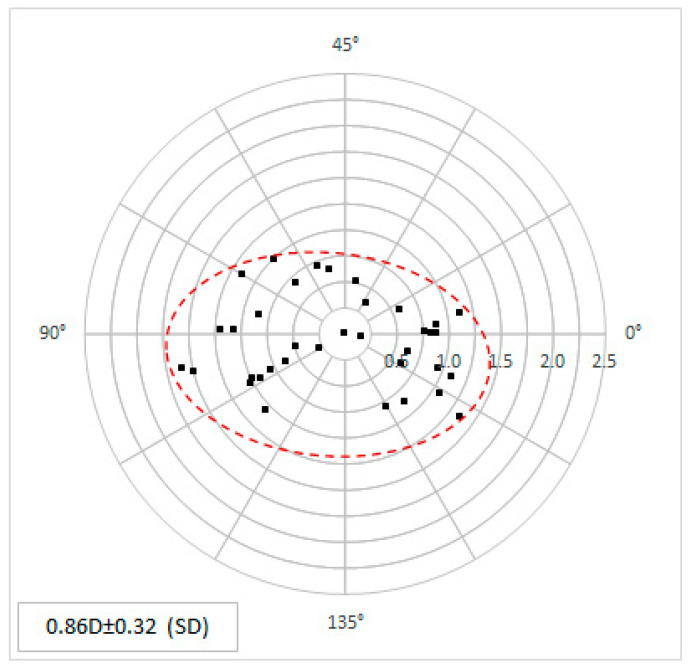
The postoperative topographic astigmatism values in power and angle are shown as double-angle plots. Each point represents the individual corneal data.

**Figure 7 micromachines-14-01009-f007:**
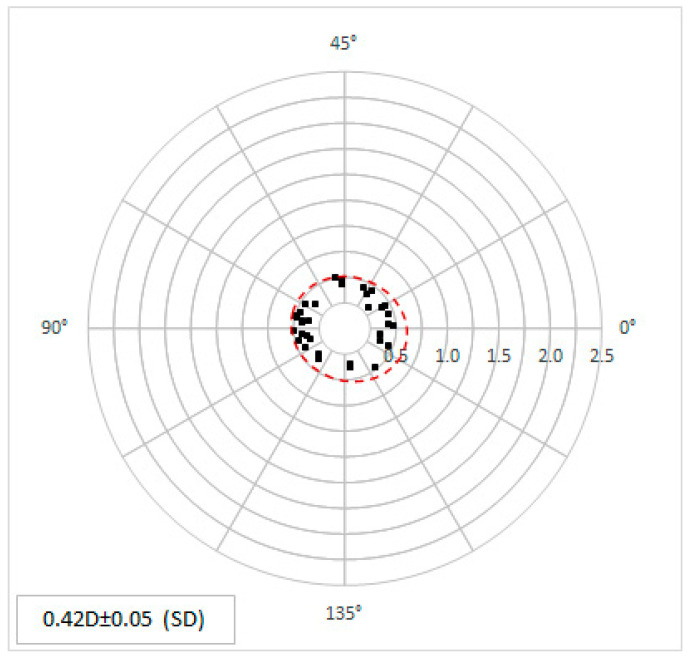
The predicted, calculated topographic astigmatism values are shown in magnitude and angle as double-angle plots. Each point represents the individual corneal data.

**Figure 8 micromachines-14-01009-f008:**
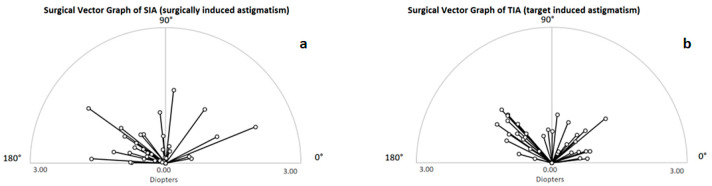
The corneal astigmatism in SIA (**a**) and TIA (**b**) is shown in a polar diagram. The astigmatism power as well as the astigmatism angle is shown. In the figure, each individual calculated arcuate incision for TIA is displayed and all postoperative measured data for SIA is shown.

**Figure 9 micromachines-14-01009-f009:**
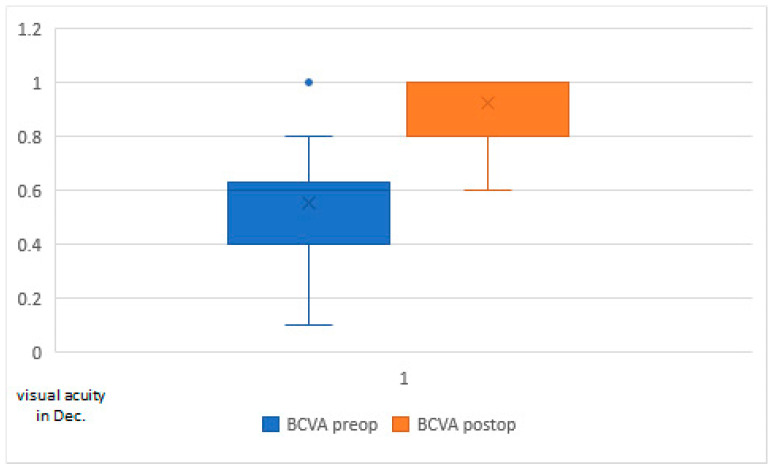
Best-corrected visual acuity (BCVA) preoperatively versus postoperatively.

## Data Availability

The data presented in this study are available upon request from the first and corresponding authors. Specifically, the datasets are archived in the clinics where the patients were treated. The data are not publicly available as they contain information that could compromise the privacy of the participants.

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
