# Peer review of "Predictability of Astigmatism Correction by Arcuate Incisions with a Femtosecond Laser Using the Gaussian Approximation Calculation"

_micromachines, 2023, doi:10.3390/mi14051009_

Round 1

Reviewer 1 Report

Authors report on results of the retrospective analysis of the postoperative astigmatism values after femtosecond laser applied arcuate corneal incisions during the cataract surgery. The manuscript is well written; it requires some corrections mentioned below.

Line 290. “Both the preoperative values and the achieved postoperative values were significant to the calculated values (p<0.001).” Suggest correcting to “significantly different”.

Line 293. Suggest defining or at least a precise description (how it is calculated, to what exactly it refers to) of Target-induced astigmatism (TIA), but also of Surgicaly-induced astigmatism (SIA)

Line 293. “Furthermore, it can be plotted what  we got, i.e. the surgical induced astigmatism (SIA).” Unclear, please rephrase.

Figure 8. It requires more explanation, and rephrasing. It is not clear what “Each arcuate incision is shown as a white dot on the diagram. The distance from the center represents the  postoperative astigmatism.” Exactly means.

Line 309. “There is no significant difference if the arcuate incisions astigmatism correction was performed asymmetrically…” Suggest: “There is no significant difference in postoperative astigmatism values between the eyes where incisions astigmatism correction was performed asymmetrically, versus…”

Figure 9. BCVA preoperatively versus postoperatively – the improvement is due to cataract-IOL exchange and not due to astigmatism alterations/improvement; in the present context, it needs specific mentioning.

Line 321. “In other words, the target refraction was achieved significantly.” Suggest omitting this sentence.

Line 324. “In our study, we show a new approach to optimize the parameters for performing arcuate incisions using femtosecond lasers. In particular, the significantly improved analysis and reproducibility of topography measurements in combination with the highly precise application of the femtosecond laser in the formation of arcuate incisions can potentially lead to better results.” An important downside of this retrospective design is the lack of control group. We don’t know exactly in comparison to what is this approach “new”, “significantly improved” compared to which method etc. This paragraph is in need of rephrasing.

Line 341. “…but we hypothesized that customized preoperative biomechanical modelling can further improve the surgical outcome”. See previous comment.

Line 375. “This supports our study as a treatment basis to perform patient-specific simulations.” Needs rephrasing.

Line 378. “Nevertheless, our work is meaningful.” This sentence is not necessary, if left standalone like it is.

Line 382. “This has the advantage, however, compared to studies where the nomograms were used as the basis for the treatment, that we did not find any overcorrections in our work, which are usually not always well tolerated by the patients.” Suggest: “Compared to studies where the nomograms were used as the basis for the treatment, treatment according to the protocol in our study produced no overcorrections, which are not well tolerated by the patients.”

End of comments.

/

Author Response

Authors report on results of the retrospective analysis of the postoperative astigmatism values after femtosecond laser applied arcuate corneal incisions during the cataract surgery. The manuscript is well written; it requires some corrections mentioned below.

Thank you very much for the very valuable review. We will address it point by point and make the necessary corrections in the paper.

Line 290. “Both the preoperative values and the achieved postoperative values were significant to the calculated values (p<0.001).” Suggest correcting to “significantly different”.

Done, thanks.

Line 293. Suggest defining or at least a precise description (how it is calculated, to what exactly it refers to) of Target-induced astigmatism (TIA), but also of Surgicaly-induced astigmatism (SIA)

Thank you very much for this input and the opportunity to comment more precisely. We have described it in detail and included it directly in the paper. This has made the work better.

Line 293. “Furthermore, it can be plotted what  we got, i.e. the surgical induced astigmatism (SIA).” Unclear, please rephrase.

Thank you very much for this important advice. We have revised the wording of this sentence and made it more precise.

Figure 8. It requires more explanation, and rephrasing. It is not clear what “Each arcuate incision is shown as a white dot on the diagram. The distance from the center represents the  postoperative astigmatism.” Exactly means.

Thank you very much for the very important advice. We have reformulated, expanded and clarified the description. It has thus become clearer.

Line 309. “There is no significant difference if the arcuate incisions astigmatism correction was performed asymmetrically…” Suggest: “There is no significant difference in postoperative astigmatism values between the eyes where incisions astigmatism correction was performed asymmetrically, versus…”

Thank you very much for your suggestion, which we were very happy to implement. It has become much better this way.

Figure 9. BCVA preoperatively versus postoperatively – the improvement is due to cataract-IOL exchange and not due to astigmatism alterations/improvement; in the present context, it needs specific mentioning.

We have expanded our work to include this note and have written a section in the discussion correspondingly. The correction is marked accordingly. Thank you very much.

Line 321. “In other words, the target refraction was achieved significantly.” Suggest omitting this sentence.

Thank you very much, we have deleted this sentence.

Line 324. “In our study, we show a new approach to optimize the parameters for performing arcuate incisions using femtosecond lasers. In particular, the significantly improved analysis and reproducibility of topography measurements in combination with the highly precise application of the femtosecond laser in the formation of arcuate incisions can potentially lead to better results.” An important downside of this retrospective design is the lack of control group. We don’t know exactly in comparison to what is this approach “new”, “significantly improved” compared to which method etc. This paragraph is in need of rephrasing.

We have significantly improved this section on the guidance of the reviewer. We have added a missing control group as a drawback of our study further down in the discussion. Thank you very much for this valuable advice.

Line 341. “…but we hypothesized that customized preoperative biomechanical modelling can further improve the surgical outcome”. See previous comment.

We have made a corresponding comment and clarification. Thank you very much.

Line 375. “This supports our study as a treatment basis to perform patient-specific simulations.” Needs rephrasing.

Thank you, we have reworded the sentence to make it clearer.

Line 378. “Nevertheless, our work is meaningful.” This sentence is not necessary, if left standalone like it is.

We completely share your opinion. We have deleted the sentence. Thank you very much.

Line 382. “This has the advantage, however, compared to studies where the nomograms were used as the basis for the treatment, that we did not find any overcorrections in our work, which are usually not always well tolerated by the patients.” Suggest: “Compared to studies where the nomograms were used as the basis for the treatment, treatment according to the protocol in our study produced no overcorrections, which are not well tolerated by the patients.”

Thank you very much for the rewording. We were very happy to adopt it. It has made the work much better.

End of comments.

Many thanks to the reviewer for the many suggestions and corrections.

Reviewer 2 Report

 Isabel Llopis and co-workers aims to address the important challenge in astigmatism correction and improve the compliance. Authors determined the numerical predictability of astigmatism correction with a femtosecond laser using an optimized algorithm in a customized application. Authors relied on biomechanical models and Gaussian approximation curve calculations for surgical planning. Authors conclude that customized simulations based on corneal biomechanics are a valuable tool for correcting mild astigmatism with corneal incisions in cataract surgery. It will thus facilitate Postoperative visual outcomes. Authors discussed the limitations and strengths of their study.

The objective, datasets and discussions are clear and deserved to be published. Statistics are also fine. However, they are certain areas for improvement before final publication.

Comments for authors:

1.     Figures are represented very brief sometimes just one line. Please rectify this. Think in readers perspective. Figures should explain themselves in standalone manner. Also, reader shouldn’t guess what segment what is. Add A and B for all panels wherever applicable.

Arial 12-14 points are good for aesthetics. Authors can use their discretion.

For example: in figure 9 what is Y axis unit? Even if unitless you better mention it. Also, which part is blue and orange? Also, add the description of statistics right below the figure in legends.

Improve this part for all figures. This part in the current part is rather weak.

See example: https://www.nature.com/articles/s41586-023-06009-4

2. Sections 2 and 3 deserve subheadings.

Maintain the connection between the methods in section 2 headings and results in 3 headings.

3.     Add scale for figure 1-4 if applicable.

4.     “Nevertheless, our work is meaningful.” Add a continuation why or delete that part. 378 line.

Author Response

Isabel Llopis and co-workers aims to address the important challenge in astigmatism correction and improve the compliance. Authors determined the numerical predictability of astigmatism correction with a femtosecond laser using an optimized algorithm in a customized application. Authors relied on biomechanical models and Gaussian approximation curve calculations for surgical planning. Authors conclude that customized simulations based on corneal biomechanics are a valuable tool for correcting mild astigmatism with corneal incisions in cataract surgery. It will thus facilitate Postoperative visual outcomes. Authors discussed the limitations and strengths of their study.

The objective, datasets and discussions are clear and deserved to be published. Statistics are also fine. However, they are certain areas for improvement before final publication.

Many thanks to the reviewer and his assessment, especially we look forward to his input which has made our work better.

Comments for authors:

  1. Figures are represented very brief sometimes just one line. Please rectify this. Think in readers perspective. Figures should explain themselves in standalone manner. Also, reader shouldn’t guess what segment what is. Add A and B for all panels wherever applicable.

Arial 12-14 points are good for aesthetics. Authors can use their discretion.

For example: in figure 9 what is Y axis unit? Even if unitless you better mention it. Also, which part is blue and orange? Also, add the description of statistics right below the figure in legends.

Improve this part for all figures. This part in the current part is rather weak.

See example: https://www.nature.com/articles/s41586-023-06009-4

Thank you very much for the extraordinarily important and good advice. I have clarified and significantly expanded the text of the figures. Furthermore, figures 8 and 9 have been revised with the points you noted. As far as the choice of typeface and size are concerned, these are fixed by the publisher, which I have adopted in full. With this revision the work has become much better.

  1. Sections 2 and 3 deserve subheadings.

Maintain the connection between the methods in section 2 headings and results in 3 headings.

We have followed her recommendation and have placed appropriate subheadings in both sections 2 and 3. This has made it clearer. Thank you very much for this input.

  1. Add scale for figure 1-4 if applicable.

Where applicable, we have made appropriate comments. Thank you very much.

  1. “Nevertheless, our work is meaningful.” Add a continuation why or delete that part. 378 line.

We completely share your opinion. We have deleted the sentence and dropped it. Thank you very much.

Round 2

Reviewer 2 Report

The authors addressed my concerns and revised the script appropriately.